# Perinate and eggs of a giant caenagnathid dinosaur from the Late Cretaceous of central China

Hanyong Pu[1], Darla K. Zelenitsky[2], Junchang Lü[3], Philip J. Currie[4], Kenneth Carpenter[5], Li Xu[1], Eva B. Koppelhus[4], Songhai Jia[1], Le Xiao[1], Huali Chuang[1], Tianran Li[1], Martin Kundrát[6] & Caizhi Shen[3]

The abundance of dinosaur eggs in Upper Cretaceous strata of Henan Province, China led to the collection and export of countless such fossils. One of these specimens, recently repatriated to China, is a partial clutch of large dinosaur eggs (*Macroelongatoolithus*) with a closely associated small theropod skeleton. Here we identify the specimen as an embryo and eggs of a new, large caenagnathid oviraptorosaur, *Beibeilong sinensis*. This specimen is the first known association between skeletal remains and eggs of caenagnathids. Caenagnathids and oviraptorids share similarities in their eggs and clutches, although the eggs of *Beibeilong* are significantly larger than those of oviraptorids and indicate an adult body size comparable to a gigantic caenagnathid. An abundance of *Macroelongatoolithus* eggs reported from Asia and North America contrasts with the dearth of giant caenagnathid skeletal remains. Regardless, the large caenagnathid-*Macroelongatoolithus* association revealed here suggests these dinosaurs were relatively common during the early Late Cretaceous.

[1] Henan Geological Museum, Zhengzhou 450016, China. [2] Department of Geoscience, University of Calgary, Calgary, Alberta, Canada T2N 1N4. [3] Institute of Geology, Chinese Academy of Geological Sciences, Beijing 100037, China. [4] Department of Biological Sciences, University of Alberta, Edmonton, Alberta, Canada T6G 2E9. [5] Prehistoric Museum, Utah State University, 155 East Main Street, Price, Utah 84501, USA. [6] Center for Interdisciplinary Biosciences, Faculty of Science, University of Pavol Jozef Safarik, Kosice 04154, Slovak Republic. Correspondence and requests for materials should be addressed to D.K.Z. (email: dkzeleni@ucalgary.ca) or to J.L. (email: Lujc2008@126.com).

In the late 1980s and early 1990s, thousands of dinosaur eggs were excavated and collected by local farmers from Cretaceous rocks of Henan, China[1]. During this time, China struggled to control the export of these eggs, many of which were sold overseas in rock and gem shows, stores, and markets by the early 1990s. Some of these exported eggs were prepared in other countries and revealed beautifully preserved embryos[2].

Of the numerous specimens exported from China, one of the most significant turned out to be a small skeleton associated with a partial clutch of the largest known type of dinosaur egg (that is, *Macroelongatoolithus*). The unprepared specimen was imported into the USA in mid-1993 by The Stone Company, which exposed the skeleton and eggs during preparation (Supplementary Fig. 1). The specimen was featured in a cover article for National Geographic Magazine[2], and the skeleton became popularly known as 'Baby Louie' in recognition of Louis Psihoyos, the photographer for the article.

In 2001, Baby Louie was acquired from the Stone Company by the Indianapolis Children's Museum where it was put on public exhibit for 12 years. The museum's intention was to repatriate the specimen to China, but an agreement for its return was not finalized until 2013. In December 2013, twenty years after the specimen was collected, Baby Louie found its final home in its province of origin at the Henan Geological Museum[3].

Although Baby Louie was known to have been collected in Henan Province, the exact locality was uncertain when the specimen was imported into the United States. However, it has since been learned that it was collected by Mr Zhang Fengchen in the Xixia Basin in western Henan between December 1992 and early 1993 (ref. 3). In February 2015, led by one of farmers who participated in the original excavation, five of the authors (Pu, Currie, Lü, Koppelhus and Jia) visited the fossil locality (Supplementary Fig. 2) and found egg fragments that are identical to those of the Baby Louie specimen. The specimen is only a small section of what would have likely been a large, ring-shaped nest of *Macroelongatoolithus* eggs; other blocks from the same nest were probably also sold outside of China.

Baby Louie is the only reported skeleton found in close association with *Macroelongatoolithus* eggs, and has been a key to the identity of these large eggs for many years. Initial attempts by certain authors (Carpenter, Currie, Zelenitsky) to identify the specimen in 1994–1995 revealed that the eggs were morphologically similar to, although considerably larger than those known for oviraptorids[4]. Although the identity of the skeleton was initially perplexing due to its uniqueness and preservation, both the skeleton and the eggs were identified as oviraptorosaurian by the late 1990s. However, study of this specimen was halted until its repatriation in 2013 because of ongoing concerns related to its legality outside China.

Oviraptorosauria comprises a group of Cretaceous maniraptoran theropods found in Asia and North America[5]. In China, most species belong to the clade Oviraptoridae, except for basal early Cretaceous forms such as *Incisivosaurus*[6] and *Caudipteryx*[7] and the Late Cretaceous, large caenagnathid *Gigantoraptor*[8]. Oviraptorosaurs are geographically distributed in northern[7,9–12], central[13,14] and southern China[15–24]. Here we describe the perinate skeleton and associated eggs of the Baby Louie specimen as the remains of a new species of large caenagnathid, from the early Upper Cretaceous Gaogou Formation of Henan, central China. This specimen represents the first known eggs closely associated with skeletal remains of a caenagnathid oviraptorosaur and provides insight into the early ontogeny, nesting and distribution of these dinosaurs.

## Results

### Systematic palaeontology.

<div align="center">

Theropoda Marsh 1881 (ref. 25).
Oviraptorosauria Barsbold 1976 (ref. 26).
Caenagnathidae Sternberg 1940 (ref. 27).
*Beibeilong sinensis* gen. et sp. nov. (Figs 1–3).

</div>

**Etymology.** The generic name is derived from Chinese Pinyin 'beibei' for baby and 'long' for dragon. The specific name is derived from Latin referring to its discovery in China.

**Holotype.** HGM 41HIII1219, a small, semi-articulated skeleton ('Baby Louie') associated with a partial nest of 6–8 eggs. The specimen is housed in the Henan Geological Museum (HGM), Zhengzhou, China.

**Locality and horizon.** The specimen was discovered at a latitude/longitude of 33°15′30″ N, 111°43′41″ E in Heimao-gou, 2 km east of Zhaoying Village, Yangcheng Township, Xixia County, Henan Province, People's Republic of China. The locality is in the Upper Cretaceous Gaogou Formation (Cenomanian—Turonian)[28,29].

**Diagnosis.** A large caenagnathid that has the following unique suite of features: antorbital fossa demarcated by sharply defined alveolar and dorsoposterior trending ridges, posterodorsal margin of lacrimal overlapped by frontal, subantorbital portion of the maxilla is inset medially, pronounced retroarticular process with a distinct concave posterior facet (roughly as tall at the base as it is wide), preacetabular process of the ilium longer than postacetabular process, posterior end of the postacetabular process truncated or broadly rounded, and accessory trochanter of the femur weakly developed.

**Description and comparisons**. HGM 41HIII1219 is a small, semi-articulated skeleton (Baby Louie) associated with a partial nest of 6–8 eggs (Figs 1 and 2; Supplementary Note 1). The skeleton is lying on top of the clutch on its left side across two adjacent eggs (Fig. 1). The perinate animal was outside the confines of any egg when it was buried, although it is closely associated with the eggs and isolated eggshell fragments. Four elongate eggs are present in the upper layer of the specimen and the ends of two eggs are visible in a lower layer, 2.5–6 cm below the upper eggs. The eggs are incomplete and compressed. Compression makes it difficult

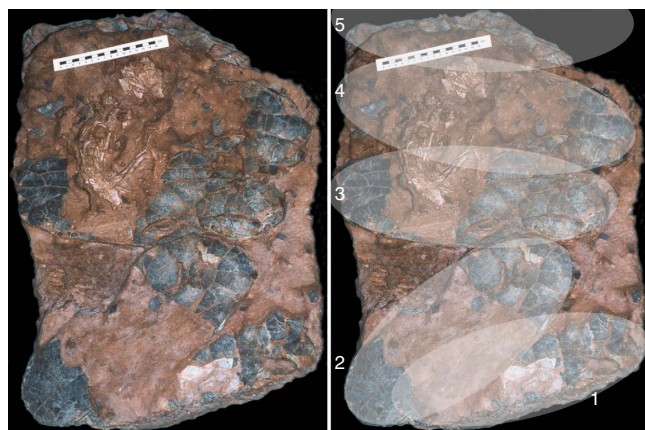

**Figure 1 | Photograph of eggs and skeleton of *Beibeilong sinensis* (HGM 41HIII1219).** Right image shows schematic overlay of approximate locations of individual eggs. Eggs 1 through 4 are in an upper layer just beneath the skeleton, whereas Egg 5 is in a lower layer of the block. Scale bar is in centimetre.

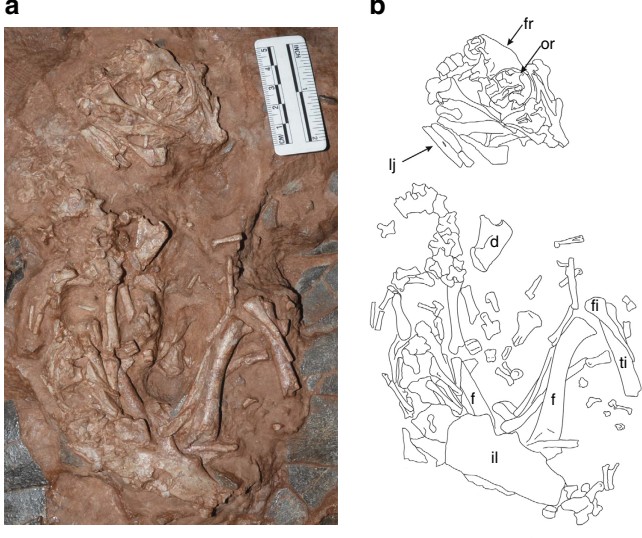

**Figure 2 | Skeleton of _Beibeilong sinensis_ (HGM 41HIII1219).**
(**a**) Photograph. (**b**) Highly schematic outline shows general layout of the skeleton (illustrated by Zhaochuang). Scale bar, 5 cm. d, dentary; f, femur; fi, fibula; fr, frontal; lj, lower jaw; or, orbit; ti, tibia.

to assess the original dimensions or determine if the eggs have a pointed pole as described for some oviraptorosaurs[18]. The measured dimensions of the two most complete eggs are 400 and 450 mm in length, and are ~150 mm across the equatorial regions (Supplementary Table 1). The eggs are subhorizontally positioned, with eggs in the lower level slightly inclined relative to eggs in the upper layer. The long axes of the eggs trend in the same general direction, although converge at one end towards what would have been the center of the nest (Fig. 1). Their arrangement indicates that HGM 41HIII1219 was a small part of a large ring-shaped nest. The subpolar and equatorial regions of the eggs show linearituberculate ornamentation, whereas the polar regions are smooth or weakly textured (Fig. 4a). The eggshell ranges from 1.70 to 2.56 mm ($n = 15$ measurements) in total thickness. Histologically, the eggshell is a two-layered structure with an outer continuous layer and inner mammillary layer delimited by an abrupt, undulatory boundary (Fig. 4b). The eggs in HGM 41HIII1219 are similar in histology, texture, shape, and arrangement to other elongatoolithids described for oviraptorids[4,9,30], although they are well over twice the length. The large size and shell thickness identifies them as the oospecies _Macroelongatoolithus xixiaensis_ (oofamily Elongatoolithidae), eggs and nests of which have been reported previously from Henan[29,31,32].

As preserved, Baby Louie had a snout to vent length of about 38 cm (Supplementary Table 2). All bones of the skeleton show fibrous juvenile bone texturing. Most of the forelimbs, feet and tail are missing or not visible.

The skull has collapsed into a layer of overlapping bones, with bones primarily from the right side visible (Fig. 3a,b). The skull is 66 mm long as preserved, and the orbit is large and round. Most of the right premaxilla is missing although two fragments adhere to the anterior margin of the nasal, one of which reveals it has a sculptured external surface. The right maxilla, located anterior to the lacrimal and jugal, lacks teeth and alveoli. It is relatively short and low, and forms the anteroventral margin of a small antorbital fenestra. The maxilla has a pronounced anteroposterior shelf below the antorbital fossa, which is continuous anteriorly with a dorsoposteriorly trending ridge on the external surface of the maxilla (Fig. 3a,b). The two ridges demarcate the antorbital

fossa. The maxilla extends a short distance ventromedially from the ridge below the antorbital fossa, but this palatal shelf is broken off anteromedially. The nasals, visible in ventral view, are fused posteriorly between the narial openings; the suture is visible anteriorly, which is similar to the immature individual of the oviraptorid _Yulong_[14]. Four aligned nutrient foramina penetrate the outside margin of the right nasal. The right lacrimal is an open crescentic bone that forms the anterior margin of the orbit. A depression on the dorsal edge of the posterior process of the lacrimal is the contact for the frontal, indicating that the frontal overlaps this bone. The antorbital (vertical) process of the lacrimal is concave posteriorly. A lacrimal foramen extends into the bone from this concavity near mid-height. The anterolateral margin of the orbit is a sharply defined ridge as in other oviraptorosaurs. The right jugal has a suborbital process that is long and low, similar to those of most oviraptorosaurs, and tapers anteriorly to contact the maxilla and lacrimal (Fig. 3b). The triangular postorbital process is a short, posterodorsally oriented projection. The frontal is highly domed and both the interfrontal and frontoparietal sutures are visible. The postorbital process of the frontal has a deep, triangular groove on its dorsal surface that broadens ventrolaterally. The incomplete right postorbital is a slender, arcuate bone that extends between the frontal and posterodorsal process of the jugal. The dorsal process of the quadratojugal is tall and slender as in oviraptorosaurs. It expands ventrally, but the jugal process was broken and presumably lost. The right quadrate has a shallow notch that separates the lateral and medial condyles, and the posterior surface is shallowly concave in lateral view. The triradiate right ectopterygoid (Fig. 3a,b), visible in the right orbit, has a well-developed pneumatic ectopterygoid fossa on the dorsal side. Other bones, probably from the palate and braincase, are present in the orbit but are difficult to identify.

Bones present from the lower jaw include the dentary, surangular, angular, and articular (Fig. 3a,b). The dentary has a general morphology similar to those of other oviraptorosaurs, particularly those specimens in which the dentaries are unfused[7,33–35]. The two dentaries are not tightly sutured or fused at the symphysis in HGM 41HIII1219, and most of the left dentary is exposed in the region of the chest in lingual view (Fig. 2). The left dentary is relatively short (27 mm along the ventral margin, or 32 mm as restored in Fig. 3c) and deep (18 mm at the anterior margin of the external mandibular fenestra), with proportions that are most similar to _Gigantoraptor_ and _Microvenator_ (Supplementary Note 2). The dentary is thin and plate like, and as expected for oviraptorosaurs it has a sharp edentulous anterodorsal margin. Anteriorly, it curves medially to form a robust, steeply sloping symphysis (Fig. 3c). The symphysial region is downturned in relation to the ventral margin of the dentary. The symphysis has a rugose, anterodorsally inclined contact. The posterior edge of the left dentary is deeply emarginated due to a large external mandibular fenestra. A ridge on the lingual surface extends from midheight of the symphysis to the posterodorsal process, forming a shelf below the cutting edge of the jaw margin anterodorsally (Fig. 3c). A much weaker ridge extends along the ventral margin of the dentary to the posteroventral process, and above this is a shallow Meckelian groove as in other oviraptorosaurs[34,36,37]. The anterior terminus of the groove is above the ventral ridge as in _Microvenator_ and oviraptorids[34] rather than below the symphysis as in most caenagnathids[36]. Although most of both the posterodorsal and posteroventral processes are missing on the left dentary, the posteroventral process (as seen on the right dentary) probably extended farther posteriorly than the posterodorsal process. The right surangular is complete, although the anterior tip is covered by the right maxilla

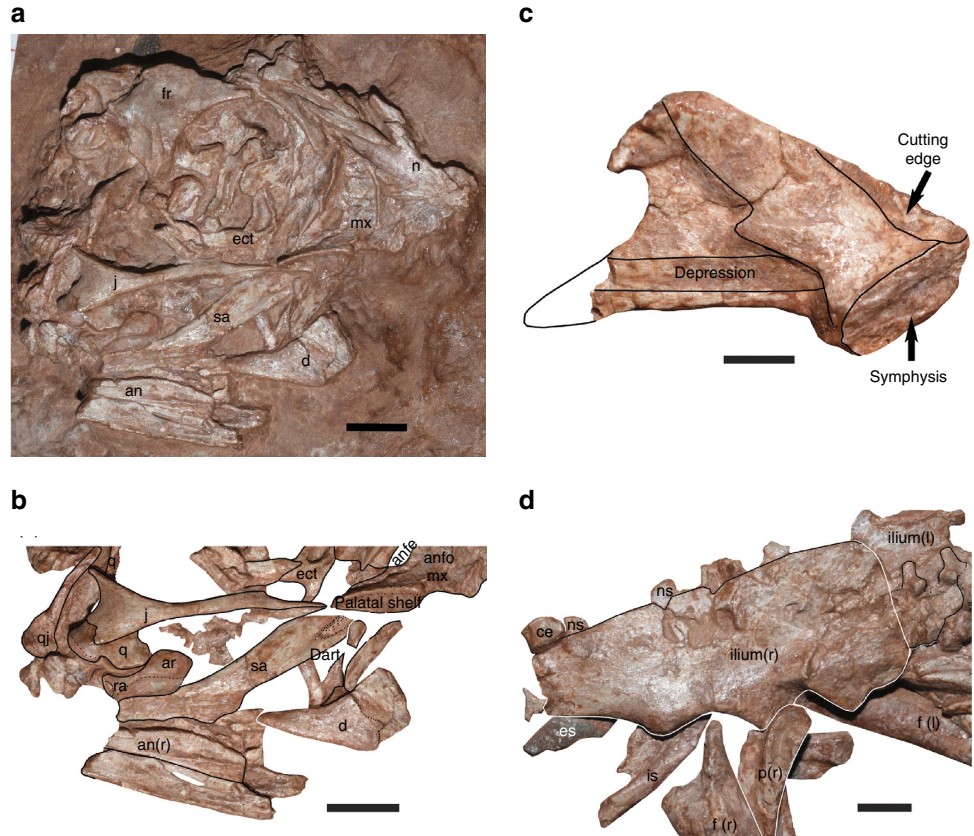

**Figure 3 | Photographs of *Beibeilong sinensis* (HGM 41HIII1219).** (**a**) Skull as preserved on right side. Scale bar, 1 cm. (**b**) Bones identified in lower part of skull and mandibles from (**a**). Scale bar is 15 mm. (**c**) Left dentary in medial view. Posteroventral process restored from the partial right dentary. Scale bar is 5 mm. (**d**) Pelvic region in right lateral view. Ilium shows juvenile bone texture, and association with sacral spines and last few dorsals. Scale bar, 1 cm. an, angular; anfe, antorbital fenestra; anfo, antorbital fossa; ar, articular; ce, centrum; d, dentary; dart, dentary articulation; ect, ectopterygoid; es, eggshell; f, femur; fr, frontal; is, ischium; j, jugal; max, maxilla; n, nasal; ns, neural spine; p, pubis; q, quadrate; qj, quadratojugal; ra, retroarticular process; sa, surangular.

(Fig. 3b). The lateral surface of the surangular has a posteriorly tapering depression for articulation with the posterodorsal process of the dentary. Like *Gigantoraptor*[8], it appears to lack a surangular spine. The dorsal margin curves posteroventrally from a relatively low coronoid prominence. The articular is not fused with the surangular. It has a pronounced convex crest-like articulation for the quadrate, typical of oviraptorids. The crest is noticeably longer anteroposteriorly than the quadrate condyles. The articular extends posteriorly into a short but pronounced retroarticular process that has a distinct concave posterior facet.

Most regions of the vertebral column are represented by about a dozen vertebrae that are generally incomplete and disarticulated (Fig. 2). An anterior cervical neural arch is visible in dorsal view, has a low neural spine, and an X-shaped dorsal profile like other oviraptorosaurs[34]. An isolated centrum in the region has parapophysis relatively high on the centrum, which suggests that it represents an anterior cervicodorsal vertebra. The last dorsal centrum is associated with its neural arch in front of the first sacral, and appears to have a large pleurocoel in its side. Although mostly covered by the right ilium, six sacrals are likely present based on the number of neural spines protruding above the dorsal edge of the ilium and on the spacing where spines are not visible.

Few elements of the pectoral girdle and forelimbs are identifiable (Fig. 2). The right scapula is strap-like and covered anteriorly by vertebrae of the presacral region, and a short fragment of the left scapula lies adjacent to it. In the same region, one curved piece may represent a furculum attached to an acromion fragment of the scapula. Two limb shafts largely covered by other bones may belong to the humeri. A complete ulna (L = 29 mm) and partial radius are associated with another egg (#4) in the specimen, which may belong to the skeleton or to another perinate (Supplementary Note 1). The olecranon process of this ulna is poorly developed as in other embryonic oviraptorids[38].

All bones of the pelvic girdle are represented (Fig. 3d). The right ilium is complete and 66 mm long. The preacetabular ala is about 1.5 times longer than the postacetabular blade. This proportion is similar to those of *Chirostenotes*, *Microvenator*, *Nankangia*, *Nomingia* and *Rinchenia*, whereas most oviraptorid dinosaurs have ilia with preacetabular and postacetabular lengths that are subequal. In lateral view, the ilium has a straight to gently convex dorsal margin, and the postacetabular process is squared off (Fig. 3d). The preacetabular process curves anteroventrally from the pubic peduncle, and is deep relative to the postacetabular process. The acetabulum forms a broad gentle arch, but there is no evidence of a supra-acetabular shelf. The right pubis is in contact with the pubic peduncle of the ilium (Fig. 3d), and inclines ventroposteriorly. The proximal part of the shaft extends along the anterior margin of the right femur. Most of the shaft is covered by the femur, and emerges distally for a short distance behind the distal half of the femur where it is broken. The left pubis lies directly under the right one, and extends most of length of the femur before it becomes covered.

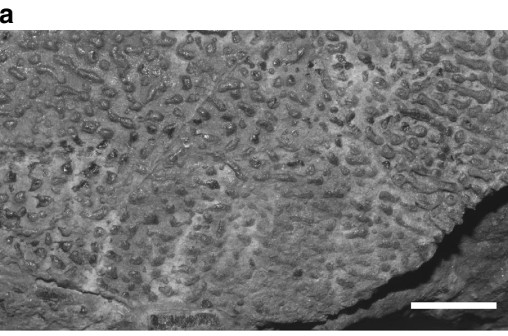

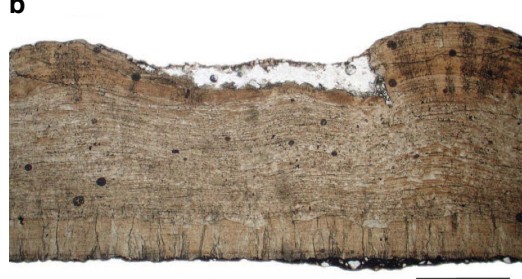

**Figure 4 | Eggshell associated with *Beibeilong sinensis* (HGM 41HIII1219).** (**a**) Photograph showing the ornamentation on the outer surface of *Macroelongatoolithus* eggshell. Scale bar, 2 cm. (**b**) Photomicrograph showing histological section of *Macroelongatoolithus* eggshell in radial view shows two microstructural layers (lower mammillary layer and upper continuous layer) demarcated by an undulatory boundary (arrow). Scale bar is 1 mm.

The left pubis exposes the pubic apron on the medial surface of the distal quarter. The pubes were not coossified at the pubic apron, representing another juvenile trait. The right ischium is displaced from the peduncle, and the rod-like shaft projects posteroventrally from the acetabulum. The proximal end of the ischium is underneath the ilium whereas the distal end is missing. A small portion of the obturator process is visible, but the ischium has rotated so it seems to be dorsal to the shaft (Fig. 3d).

The hindlimbs are represented by several elements (Fig. 2), although most bones of the foot are missing. The left femur projects anteriorly from beneath the ilium (Fig. 3d), and ribs cover the distal end. The right femur is exposed in lateral view and is articulated with the acetabulum and tibia/fibula. The ends of the right femur are incomplete. The proximal end extends into the acetabulum but appears to have been largely destroyed by an insect boring, whereas the distal end would have been capped by a cartilaginous extension of the bone. As preserved, the right femur is 66.8 mm long. However, the distance between the top of the acetabulum and the head of the tibia is 75 mm, which is probably a closer approximation of the actual length of the femur. The femur is robust and slightly bowed (Fig. 2). Like *Gigantoraptor* but unlike other oviraptorosaurs[39], there is no evidence of a ridge extending along the shaft between the anterior trochanter and the distal medial condyle. The distal end has a shallow posterior depression that separates the lateral and medial 'condyles', both of which are abruptly truncated in a flat surface that is not subdivided distally (Supplementary Fig. 3a). The right tibia is missing its distal portion. The cnemial crest does not project far from the shaft as in other embryonic oviraptorosaurs[33,38] (Supplementary Fig. 3b), and lacks the distinct boss that is present on the distal end of the crest in *Citipati*, *Gigantoraptor* and *Nomingia*[39]. The lateral (fibular) condyle is low and poorly defined, but continues distally in

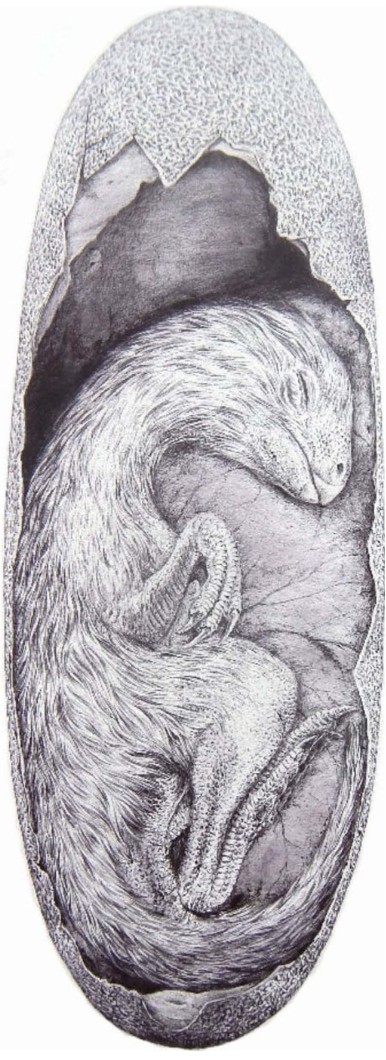

**Figure 5 | Reconstruction of *Beibeilong* embryo *in ovo*.** The drawing shows the approximate size of the *Beibeilong* embryo inside a *Macroelongatoolithus* egg (drawn by Vladimir Rimbala).

a distinct ridge that is connected with the fibular crest (Supplementary Fig. 3b). The fibular crest is low as in *Gigantoraptor*[8], but has parallel, proximolaterally extending ridges for the attachment of ligaments. The shaft of the tibia is robust. The fibula had slid a short distance up the tibia, and is missing its distal half. Approximately 22 mm from the proximal end is a shallow depression that bounds a more distal moundlike, anterolaterally projecting iliofibularis tubercle (Supplementary Fig. 3b). Six disarticulated pedal phalanges in the region of the right knee probably represent II-1, III-2, both right and left IV-2, IV-3 and IV-4.

**Maturity and growth.** The size, posture, and other features of the skeleton indicate the animal was immature (likely embryonic) at the time of death even though it is not preserved within the confines of an egg. The large orbit, domed frontals, lack of fused skull bones, ill-defined or undeveloped processes of long bones, unfused neural arches and highly vascularized bone texture of the skeleton are typical of embryonic or immature individuals[33,38,40]. The chin of *Beibeilong* is tucked down towards the chest in a posture reminiscent of *in ovo* dinosaur[33], crocodile[41,42] and bird[43] embryos (Fig. 2). The curled skeleton is only 23 cm long

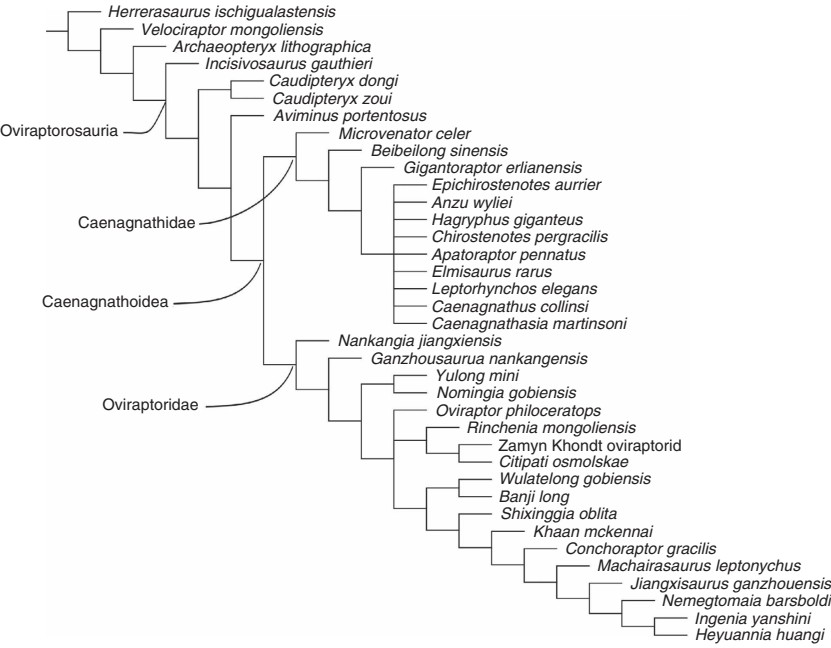

**Figure 6 | Phylogenetic position of *Beibeilong sinensis* gen. et sp. nov.** The strict consensus tree of 608 most parsimonious trees, obtained from a phylogenetic analysis of 37 taxa and 250 characters. Consistency index = 0.482. Retention index = 0.667.

(as measured from the top of the skull to the base of the tail) and would readily fit inside an egg over 40 cm long, probably occupying no more than two-thirds of the egg volume (Fig. 5; Supplementary Fig. 4). Given that the skeleton is not preserved within an egg and that its orientation is inconsistent with that of the eggs, it is likely that the embryo was forcefully extruded or removed from one of the underlying eggs to its current position.

Generalizations relevant to growth in large caenagnathids can be made by comparing the *Beibeilong* embryo to the adult specimen of *Gigantoraptor*. In *Beibeilong*, the dentaries are not fused at the mandibular symphysis (like in the small, juvenile caenagnathid *Microvenator*) whereas they are completely fused in *Gigantoraptor* and in other adult caenagnathids, indicating that fusion of the dentaries probably occurred post-hatching. Similarities in various proportions of the dentary between *Beibeilong* and *Gigantoraptor* suggest that the overall shape of the mandible remained relatively consistent in large caenagnathids during growth (Supplementary Note 2).The mandible-to-femur length ratio is high (~0.87) in *Beibeilong* but low in *Gigantoraptor* (0.45)[8], reflecting a considerable difference in relative skull size between the two taxa. However, a large difference in such ratios between individuals of drastically different ontogenetic stages within a species is also not unexpected through ontogeny[44].

**Phylogenetic analysis.** A phylogenetic analysis was conducted in order to determine the position of *Beibeilong sinensis* gen. et sp. nov. within Oviraptorosauria using the character matrix of Funston and Currie[45]. On the basis of 37 taxa (incl. *Herrerasaurus*, *Velociraptor* and *Archaeopteryx* as outgroups) and 250 characters (Supplementary Table 4), 608 most parsimonious trees were recovered where a strict consensus tree (Fig. 6) supports the monophyly of several clades within Oviraptorosauria, including Caudipterygidae, Caenagnathoidea, Oviraptoridae and Caenagnathidae as in recent analyses[45,46]. *Beibeilong sinensis* is nested within Caenagnathidae, and is more derived than *Microvenator celer* but is basal to *Gigantoraptor erlianensis*. Its phylogenetic position is

consistent regardless of whether or not the characters we consider as ontogenetically variable (that is, characters 82, 84, 162, 165, 166, 184, 185, 187, 222, 224 and 248) are coded for *Beibeilong sinensis*. Exclusion of such characters (that is, coded with a '?') reveals three autapomorphies for the new species: subantorbital portion of the maxilla is inset medially (character 10, state 1), posterior end of the postacetabular process is truncated or broadly rounded (character 141, state 0), and accessory trochanter of the femur is weakly developed (character 204, state 0). Coding these characters recovers two additional autapomorphies for *Beibeilong* that are likely due to immaturity of the individual, including: anterodorsal margin of dentary straight in lateral view (character 84, state 0), and frontals strongly arched, projecting well above orbit in lateral view to contribute to nasal-frontal crest (character 162, state 1). The stability of the phylogenetic position of *Beibeilong* regardless of whether or not ontogenetically-variable characters are coded suggests that *Beibeilong* is more basal than *Gigantoraptor* and not an embryonic individual of this taxon.

**Discussion**

The Baby Louie specimen has been key to determination of the taxonomic affinity of the largest known dinosaur egg and nest type (*Macroelongatoolithus*) since shortly after its illegal export from China over two decades ago (Supplementary Note 3). Its description and taxonomic identification herein reveals the first known occurrence of eggs and embryonic remains of a caenagnathid oviraptorosaur. As the egg layers of *Macroelongatoolithus*, caenagnathids produced the largest known eggs (reported to have reached 61 cm in length[32]) and nests (2–3 m in diameter[47]) for dinosaurs. Although *Beibeilong* does not form a clade with the large caenagnathid *Gigantoraptor*, which has an estimated body mass of 1,400 to 3,246 kg (refs 8,48,49), the *Macroelongatoolithus* eggs associated with *Beibeilong* indicate a comparatively large adult body size. A prior estimate of adult body mass for a 42 cm-long *Macroelongatoolithus* egg was 1,100 kg (ref. 50). Large body size in oviraptorosaurs may be derived for caenagnathids as *Beibeilong*

and *Gigantoraptor* occupy more basal positions among Caenagnathidae.

Aside from the large size of the eggs associated with *Beibeilong*, caenagnathids and oviraptorids are similar with respect to characteristics of their eggs and nests, and probably their nesting behaviours. The similar clutch configuration, egg shape, as well as eggshell texturing and histology shared by these two clades suggest that these features are primitive for at least Caenagnathoidea. The arrangement of the eggs in the specimen indicate that the original nest was ring-shaped like those of *Macroelongatoolithus* and oviraptorids, a configuration that may be related to brood-like behaviours of the adults[30]. Based on complete *Macroelongatoolithus* nests (19, 26 and 33 eggs reported by Huh et al.[46]), the original clutch of the *Beibeilong* specimen would have had many more eggs than the 6–8 present in the specimen. Two layers of eggs are present, although other *Macroelongatoolithus* nests are reported to have only a single layer of eggs[32,47,51], whereas oviraptorid nests generally have two or three layers. It is evident that eggs were inclined in the *Beibeilong* nest as reported for oviraptorid nests, although it is uncertain if they dipped towards or away from the centre of the nest because the original inclination of the block to horizontal is unknown. The similarities in the arrangement of the eggs between these large clutches and those of small oviraptorids suggest a similar style of nest and incubation where the adult sat in the centre of the clutch[30]. The entire *Beibeilong* nest was likely in the latter stages of incubation when it was buried because perinate skeletal remains were found associated with 2–3 eggs in the specimen.

Given the link established here between *Macroelongatoolithus* eggs and giant caenagnathids, prior discoveries of these eggs provide insight into the distribution and occurrence of giant oviraptorosaurs, dinosaurs that are otherwise poorly represented by skeletal remains. Although giant oviraptorosaurs are known only from this specimen of *Beibeilong*, a partial skeleton of a single individual of *Gigantoraptor*[8] from China, and a pair of dentaries from Mongolia[52], many occurrences of *Macroelongatoolithus* egg remains are reported from China[29,31,32,51,53,54], Korea[47], Mongolia[55] and North America[56,57]. The geographical distribution and abundant occurrences of *Macroelongatoolithus* remains reveal that giant oviraptorosaurs were relatively widespread and perhaps even common in the early part of the Late Cretaceous, even though their skeletal remains are scarce and have yet to be identified in many regions.

## Methods

**Specimen preparation and description.** The oviraptorosaur skeleton was not exposed when the block was shipped to the United States from China (Supplementary Fig. 1). Charlie Magovern of The Stone Company (Boulder, Colorado, USA) put an estimated 700 h of work over several years into manually preparing the specimen under a microscope. The specimen was prepared in the same orientation as it was found in the field with the perinate skeleton on top of the eggs. Magovern had recognized that part of an egg had been attached onto the block (likely to increase the price of the specimen), so this piece was removed during preparation. At this time, some of the authors (Carpenter, Currie, Zelenitsky) undertook additional research to reveal specific details when they were trying to make the initial identification. Whereas the eggs were recognized as being virtually identical to those ascribed to oviraptorosaurs, albeit larger, skeletal features of these dinosaurs were not readily apparent in the perinate. A bone in the chest region was eventually identified as an unfused dentary of an oviraptorosaur by Peter Makovicky, who recognized its similarity to *Microvenator*.

For descriptive work, the skeletal and egg remains were examined using binocular microscopes and measured using digital calipers. Seven pieces of eggshell were removed during preparation for histological examination with optical and scanning electron microscopy (SEM). Two eggshell fragments were ultrasonically cleaned and mounted on aluminum stubs for SEM analysis on a FEI Quanta FEG250. Five eggshell fragments were cut into thin sections using the techniques of Hirsch and Quinn[58], for examination with a petrographic microscope (Leica DM2500P). Thin sections were observed under ordinary and polarizing light

microscopy. Digital photomicrographs were captured during microscopic analyses of the eggshell. The specimen was also photographed at various stages of preparation and study by Louis Psihoyos (National Geographic), Flo and Charlie Magovern (The Stone Company) and the authors.

**Phylogenetic analysis.** To determine the phylogenetic position of *Beibeilong sinensis* within Oviraptorosauria, this taxon was added to the phylogenetic data set of Funston and Currie[44] which had been modified from Lamanna et al.[46] via the addition of 20 characters. Our matrix consists of 250 characters and 34 oviraptorosaur taxa, plus the outgroups (*Archaeopteryx*, *Herrerasaurus* and *Velociraptor*; Supplementary Table 4). Only about one-third of the 250 characters in the matrix could be coded for *Beibeilong sinensis* based on HGM 41HIII1219 and HGM 699 (Supplementary Table 3).

The 250-character matrix was used to run two separate maximum parsimony analyses in TNT v1.1 (ref. 59) for which: (1) all possible characters were coded for *Beibeilong sinensis*; and (2) ontogenetically-variable characters for *Beibeilong sinensis* were recoded with a '?', in order to reduce the influence of juvenile characteristics on the phylogenetic hypothesis. The characters recoded as '?' for *Beibeilong sinensis* in the second analysis include characters 82, 84, 162, 165, 166, 184, 185, 187, 222, 224 and 248.

In TNT, a matrix was subjected to a maximum parsimony analysis, first conducting a 'new technology' search (with default parameters for sectorial search, ratchet, tree drift, and tree fusion) that recovered a minimum tree length in 10 replicates. This procedure aims to broadly sample tree space and identify individual tree islands. The recovered most parsimonious trees (MPTs) were then subjected to a traditional search with TBR branch swapping, which more fully explores the tree islands found in the 'new technology' search. Bremer values were used to assess clade support.

**Nomenclatural act.** This published work and the nomenclatural act it contains have been registered in ZooBank, the proposed online registration system for the International Code of Zoological Nomenclature (ICZN). The ZooBank LSID (Life Science Identifier) can be resolved and the associated information viewed through any standard web browser by appending the LSID to the prefix 'http://zoobank.org/'. The LSID for this publication is: urn:lsid:zoobank.org:pub:00CBBC39-41BC-4900-83AE-C38B5A25F970.

**Data availability.** The authors confirm that all relevant data are included in the paper and its Supplementary Information files.

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

## Acknowledgements

We thank Charlie Magovern for all the work he invested in the preparation of the specimen, and Charlie and Flo Magovern for their hospitality. Jeff Paget and Dallas Evans (Indianapolis Children's Museum) assisted with the study of the specimen when it was in Indianapolis, but more importantly facilitated its return to China. Mr Zhang Xingliao, the former director of Henan Geological Museum, and Professor Dong Zhiming of the IVPP who helped in the return of the specimen to China are also gratefully acknowledged. And finally, our thanks go to Louis Psihoyos, who is the namesake of Baby Louie, for his involvement in the project and his creative talents. J.L. was funded by the National Natural Science Foundation of China (Grant No.: 41672019; 41272022), the Fundamental Research Funds for the Chinese Academy of Geological Sciences (Grant No.: JB1504) and China Geological Survey (Grant No:DD20160201). D.K.Z. and P.J.C. were funded by NSERC Discovery Grants.

## Author contributions

J.L., P.J.C and D.K.Z designed the project. J.L., H.P., L.X., S.J., L. X, H.C., T.L., and C.S. organized the curation and preparation of the specimen. P.J.C., J.L., D.K.Z, E.K., and K.C. performed the research, with J.L., D.K.Z., and P.J.C. performing the phylogenetic analysis. D.K.Z., J.L., P.J.C., and M.K. wrote the manuscript. All authors discussed the results and provided input on the manuscript.

## Additional information

**Competing interests:** The authors declare no competing financial interests.

