## [Peer Review File · Nature Communications]

Reviewers' comments:

Reviewer #1 (Remarks to the Author):

This is a superb, well-written manuscript describing a new dinosaur species that is intimately associated with a partial nest of eggs classified parataxonomically as the oogenus *Macroelongatoolithus*. Accordingly, it finally and critically establishes the association of a widespread yet enigmatic ichnotaxon (*Macroelongatoolithus*) with a dinosaurian body fossil (a giant caenagnathid). While the specimen ("Baby Louie") has been known to palaeontologists for over two decades, it has never been formally described; the work in this paper is therefore novel, and it will be a major contribution to the fields of vertebrate palaeontology and evolutionary biology. The methodology used by the authors is well established, the text is clear and straightforward, and the conclusions are sound. I would rate the paper a 9.7 out of 10.

My only substantial quibble is the use of the nickname "Baby Louie" throughout the text: while journalists and bloggers will love this nickname, it comes across (to me, at least) as tantamount to pandering to the media attention I expect this paper will generate. I suggest that the authors restrict use of the name "Baby Louie" to the Introduction, and then use the specimen number when referring to the body fossil (see my comment of lines 96–98), e.g. the limb bones mentioned in lines 220 and 221 should be stated as possibly belonging to "HGM 41HIII1219 or to another perinate" (not to "to Baby Louie or to another perinate").

Detailed comments:

Line 4: full name of fifth author is "Ken Carpenter" (not "Carpenter Ken")

Lines 96–98: the specimen number HGM 41HIII1219 should be attached to the embryo, not to the "partial nest of 6–8 eggs" as this could be challenged by other workers that the authors have created a junior synonym of *Macroelongatoolithus carlylei*. I suggest re-wording, as follows:

Holotype. HGM 41HIII1219, a small, semi-articulated skeleton ("Baby Louie") associated with a partial nest of 6-8 eggs. The specimen is housed in the Henan Geological Museum, Zhengzhou, China.

Modify subsequent mentions of specimen number as needed.

Line 136: "*Macroelongatoolithus xixiaensis*" should be "*Macroelongatoolithus carlylei*" (the former is a junior synonym of the latter).

Line 271-273: what about the in ovo pose of chicken embryos? Not the same?

Line 286: how can a "a pneumatic premaxilla" be a diagnostic character of oviraptorosaurs when below (line 296–297) "pneumatization of the premaxilla (character 8, state 1)" is listed as a diagnostic feature of caenognathoids? (i.e. the feature diagnoses a less inclusive

group than Oviraptorosauria.)

Line 307: it is not clear to me what the authors mean by "Beibeilong sinensis is supported by four autapomorphies among caenagnathids". Do they mean that the following characters are autapomorphies of Beibeilong sinensis? or that Beibeilong sinensis shares those characters with other caenagnathids (i.e. the characters are caenagnathid synapomorphies)?

Line 327: "phylogentic" should be "phylogenetic"

Line 338: should "19-26-33" be "19, 26, and 33 eggs" ?

Line 346: "center" should be "centre"

Figure 5: taxon label "Caenagnathoidae" (left side of figure) should be "Caenagnathoidea". Also: "Zamyn Khondt oviraptorid" (bottom right in figure) should not be italicized.

Reviewer #2 (Remarks to the Author):

Review of Pu et al. "A perinate of a giant caenagnathid (Oviraptorosauria: Dinosauria) associated with the largest known dinosaur eggs from the Late Cretaceous of Henan, China" to Nature Communication. 2016-08-11

The manuscript is devoted to an exciting dinosaur fossil embryo that deserves full attention and publication.

The manuscript is complete, well written, and with solid science, but some issues should be addressed in the taxonomy. In addition, some minor changes may also improve the document.

Title: "(Dinosauria: Oviraptorosauria)" rather than "(Oviraptorosauria: Dinosauria)"

Line 101: If possible, please try to provide better chronological dates. Late Cretaceous is too broad.

Diagnosis (l. 102 to 116): separate unambiguous unique diagnostic features from the comparative diagnosis. Autapomorphies (lines 308-313) should be included in the diagnosis. Clarify which bone refers in the case of the ilium (l. 112) and femur (l.114).

If the baby is an embryo (as suggested), how to explain the lack of eggshells for that specific egg? You mention that "was probably forcefully extruded or removed from the egg". This suggestion requires better explanation and data.

Provide thin-section eggshell ultrastructure image to fig. 6

If the eggs and embryos are the same species and if the eggshell (which is a true fossil and not a imprint) was named *Macroelongatoolithus xixiaensis*, then please explain why *Macroelongatoolithus xixiaensis* is not taking the name seniority over *Beibeilong sinensis*.

Please state if the eggs are oval (different pole diameter) or true ellipsoid.

Figure captions: Please include the taxonomy and specimen numbers in the figure captions, including the crocodile species.

More comments on ontogenetical features would improve the manuscript. What ontogenetical transformations do you perceive?

The position of *Beibeilong* close but more basal than *Gigantoraptor* (from coeval layers) may be consistent to the case *Beibeilong* being, indeed, just a baby *Gigantoraptor*, where the ontogenetic condition would pull it into a more basal position. Are the differences to *Gigantoraptor* just ontogenetical? Please address that hypothesis.

Suppl. Material:

Provide specimen numbers in the tables

The wording of the "absent"/"present" state characters need revision. Too many characters states are just "absent"/"present" which is poorly informative. A proper description of the plesiomorphic and apomorphic condition is required.

For instance: "195. Surangular and angular divided by posterior extension of the external mandibular fenestra: 0 absent, 1 present"

Could be transformed to something like this: "195. Surangular and angular contact: 0, surangular and angular contact posteriorly; 1, nonexistent because are divided by posterior extension of the external mandibular fenestra."

The "absent"/"present" may contribute to confusion such as in some cases like ch. 209.

"Sternum, distinct lateral xiphoid process posterior to costal margin: 0 absent, 1 present".

The plesiomorphic condition is the shallow poorly defined (opposed to distinct) process, a medial (opposed to lateral) xiphoid process, or an anterior (opposed to posterior) to costal margin?

Other example: in "214. Surangular, distinct groove on dorsal surface: 0 present; 1 absent". The apomorphic condition is a shallow groove or in position other than dorsal? Please rewrite and clarify the characters with "absent"/"present" states.

Provide the list of synapomorphies (the list of characters suffice) for the main clades.

Good work.

Octávio Mateus

Reviewer #3 (Remarks to the Author):

Review: H. Pu, P.J. Currie, J. Lu, D.K. Zelenitsky, K. Carpenter, L. Xu, E. Koppelhus, S. Jia, H. Chuang, T. Li, M. Kundrat, C. Shen. A perinate of a giant caenagnathid (Oviraptorosauria: Dinosauria) associated with the largest known dinosaur eggs from the Late Cretaceous of Henan, China

The anatomical description of this very significant specimen is thorough and sufficient to distinguish this individual from other individuals. The manuscript, however, can be improved in two major ways in my opinion. The first is relatively straight forward: the use of an updated source matrix for the phylogenetic analysis. The second is more problematic: is it in fact a reasonable action to create a new taxon name for a perinate individual, given that it may well be difficult to assess membership in this taxon in ontogenetically-older specimens. I address each of these in turn.

PHYLOGENETIC ANALYSIS: For purposes of the phylogenetic analysis, the recent paper by Funston & Currie (2016) describing *Apatoraptor* (ref. 59 in the bibliography) provides an updated version of the source matrix used in the present manuscript. The Funston & Currie version has better resolution in both Oviraptoridae and Caenagnathidae than earlier versions, and so might help break up the massive polytomies found in the present study. Although the manuscript says that this matrix was used, *Apatoraptor* is absent in Figure 5, which leads me to think that it is actually an earlier version of the matrix that was used for this figure. (The matrix in the supplementary data does contain the new Canadian form, so this is the version that should be used.)

Additionally, should the new analysis (incorporating *Beibeilong* and the new characters mentioned in this present manuscript) have the unresolved polytomies for derived caenagnathids and oviraptorids, I would recommend providing additional consensus tree techniques (e.g., majority-rule; combinable-component, etc.). This could be used to determine if there truly is no structure within these masses of taxa, or instead if there is a shared structure in the most parsimonious trees but one or more rogue taxa have multiple different positions, resulting in strict consensus in which the real structure is obscured.

SHOULD A NEW TAXON BE ERECTED FROM A PERINATE? I can understand and appreciate the desire to assign a taxon name to this relatively complete specimen. And, given that no Gaogou Formation large-bodied caenagnathid genus and species is available to which 'Baby Louie' can be referred, a new name would need to be created to accommodate it. So there is nothing wrong per se in erecting "*Beibeilong sinensis*" for this specimen's reception.

However, this new name might be quite difficult to apply in future discoveries of post-nestling individuals (especially subadults and adults) of the same taxon, as we may lack direct testable observations to refer said individual to the holotype's identity. That is because this extremely young individual may lack the morphology of a later 'ontogimorph' of the same taxon. (For a comparison, it might be difficult to identify the autapomorphies of *Alligator mississippiensis* adults in a perinate individual). This is especially likely for issues of proportions and shapes of bones, which are likely to change as an animal grows from the size of a turkey to that of however large *Beibeilong* became.

Furthermore, the differential diagnosis for *Beibeilong* may not be applicable to distinguish it from other oviraptorosaur taxa which are known almost strictly from adult or subadult individuals. (On the other hand, it would be a useful set of observations to compare to those oviraptorosaurs known from perinates, just as we could distinguish hatchling *Alligator mississippiensis* from hatchling *Crocodylus niloticus*.) Given we do not have good ontogenetic series of any oviraptorosaur taxa, we do not yet have a good prediction of the likely trajectory of changes of shape and proportion of individual skull bones (for instance) to infer a likely shape for the adult *Beibeilong* skull.

In summary, I am not against the erection of the new name, and would not say it is an inappropriate move. However, I foresee that it might cause difficulty in the future (particularly if two sympatric adult caenagnathid taxa are found in the Gaogou). However, the authors might consider simply referring to the specimen as (alternatively) "Baby Louie" and HGM 41HIII1219, and indicate that they are awaiting discovering a hypodigm including more mature individuals before naming the taxon.

ADDITIONAL COMMENTS: A very useful aspect of this study (beyond the description of this important specimen) is the recognition of new suites of characters that unite Caenagnathoidea and Caenagnathidae.

Also, an interpretive line drawing or labeled photograph of the whole specimen might be very useful. (Ideally, a 3D CT scan of the whole would be even more useful: perhaps these are planned for a future study.)

Specific Comments and Corrections

p. 2, l. 4 Ken Carpenter's name is shown inverted.

p. 5, ll. 94-94 Technically, the Greek (and later Latin) word is Sina; this root word elements become Sin- or Sino- depending on the particular suffix. So this sentence might better read: "Sinensis", Latin for "coming from China".

p. 24, l. 521 Italicize *Chirostenotes pergracilis*

p. 26, l. 571 Italicize *Macroelongatoolithus*

Response to reviewers

Reviewers' comments:

Reviewer #1 (Remarks to the Author):

This is a superb, well-written manuscript describing a new dinosaur species that is intimately associated with a partial nest of eggs classified parataxonomically as the oogenus *Macroelongatoolithus*. Accordingly, it finally and critically establishes the association of a widespread yet enigmatic ichnotaxon (*Macroelongatoolithus*) with a dinosaurian body fossil (a giant caenagnathid). While the specimen ("Baby Louie") has been known to palaeontologists for over two decades, it has never been formally described; the work in this paper is therefore novel, and it will be a major contribution to the fields of vertebrate palaeontology and evolutionary biology. The methodology used by the authors is well established, the text is clear and straightforward, and the conclusions are sound. I would rate the paper a 9.7 out of 10.

My only substantial quibble is the use of the nickname "Baby Louie" throughout the text: while journalists and bloggers will love this nickname, it comes across (to me, at least) as tantamount to pandering to the media attention I expect this paper will generate. I suggest that the authors restrict use of the name "Baby Louie" to the Introduction, and then use the specimen number when referring to the body fossil (see my comment of lines 96 - 98), e.g. the limb bones mentioned in lines 220 and 221 should be stated as possibly belonging to "HGM 41HIII1219 or to another perinate" (not to "to Baby Louie or to another perinate").
We agree and this has been modified in the manuscript as requested.

Detailed comments:

Line 4: full name of fifth author is "Ken Carpenter" (not "Carpenter Ken")
This has been modified in the MS.

Lines 96 - 98: the specimen number HGM 41HIII1219 should be attached to the embryo, not to the "partial nest of 6 - 8 eggs" as this could be challenged by other workers that the authors have created a junior synonym of *Macroelongatoolithus carlylei*. I suggest re-wording, as follows:
Holotype. HGM 41HIII1219, a small, semi-articulated skeleton ("Baby Louie") associated with a partial nest of 6-8 eggs. The specimen is housed in the Henan Geological Museum, Zhengzhou, China. Modify subsequent mentions of specimen number as needed.

This has been modified in the MS.

Line 136: "Macroelongatoolithus xixiaensis" should be "Macroelongatoolithus carlylei" (the former is a junior synonym of the latter).

Zelenitsky et al. (2000) had previously synonymised the two aforementioned oospecies, although we are following the latter recommendation of Jin et al. (2007) who suggest that the two oospecies remain distinct and so the Baby Louie eggs are attributed to Macroelongatoolithus xixiaensi rather than M. carlylei.

Line 271-273: what about the in ovo pose of chicken embryos? Not the same? *Yes they are comparable so we incorporated additional references for birds.*

Line 286: how can a "a pneumatic premaxilla" be a diagnostic character of oviraptorosaurs when below (line 296 - 297) "pneumatization of the premaxilla (character 8, state 1)" is listed as a diagnostic feature of caenognathoids? (i. e. the feature diagnoses a less inclusive group than Oviraptorosauria.)

We have clarified and modified this in the phylogenetic section of the text with the updated phylogenetic analysis.

Line 307: it is not clear to me what the authors mean by "Beibeilong sinensis is supported by four autapomorphies among caenagnathids". Do they mean that the following characters are autapomorphies of Beibeilong sinensis? or that Beibeilong sinensis shares those characters with other caenagnathids (i. e. the characters are caenagnathid synapomorphies)?

We have clarified and modified this in the phylogenetic section of the text with the updated phylogenetic analysis.

Line 327: "phylogentic" should be "phylogenetic"

This is modified in the manuscript.

Line 338: should "19-26-33" be "19, 26, and 33 eggs" ?

This is modified in the manuscript.

Line 346: "center" should be "centre"

This is modified in the manuscript.

Figure 5: taxon label "Caenagnathoidae" (left side of figure) should be "Caenagnathoidea". Also: "Zamyn Khondt oviraptorid" (bottom right in figure) should not be italicized.

This has been modified in the figure.

Reviewer #2 (Remarks to the Author):

Review of Pu et al. "A perinate of a giant caenagnathid (Oviraptorosauria: Dinosauria) associated with the largest known dinosaur eggs from the Late Cretaceous of Henan, China" to Nature Communication. 2016-08-11

The manuscript is devoted to an exciting dinosaur fossil embryo that deserves full attention and publication.

The manuscript is complete, well written, and with solid science, but some issues should be addressed in the taxonomy. In addition, some minor changes may also improve the document.

Title: "(Dinosauria: Oviraptorosauria)" rather than "(Oviraptorosauria: Dinosauria)"

Line 101: If possible, please try to provide better chronological dates. Late Cretaceous is too broad.

This has been modified in MS by including the stage.

Diagnosis (l. 102 to 116): separate unambiguous unique diagnostic features from the comparative diagnosis. Autapomorphies (lines 308-313) should be included in the diagnosis. Clarify which bone refers in the case of the ilium (l. 112) and femur (l. 114).

If the baby is an embryo (as suggested), how to explain the lack of eggshells for that specific egg? You mention that "was probably forcefully extruded or removed from the egg". This suggestion requires better explanation and data.

This interpretation is now explained clearly and in more detail in the manuscript.

Provide thin-section eggshell ultrastructure image to fig. 6

This is now included in the figures.

If the eggs and embryos are the same species and if the eggshell (which is a true fossil and not a imprint) was named Macroelongatoolithus xixiaensis, then please explain why Macroelongatoolithus xixiaensis is not taking the name seniority over Beibeilong sinensis.

Macroelongatoolithus refers to the product of the animal, i. e., the eggs, which are classified according to a parataxonomy, and not the animal itself. We have

also made it clear that the holotype of Beibeilong is the skeleton, not the eggs.

Please state if the eggs are oval (different pole diameter) or true ellipsoid.
This has now been clarified in the description.

Figure captions: Please include the taxonomy and specimen numbers in the figure captions, including the crocodile species.
This has now been included in the manuscript.

More comments on ontogenetical features would improve the manuscript. What ontogenetical transformations do you perceive?
The position of Beibeilong close but more basal than Gigantoraptor (from coeval layers) may be consistent to the case Beibeilong being, indeed, just a baby Gigantoraptor, where the ontogenetic condition would pull it into a more basal position. Are the differences to Gigantoraptor just ontogenetical? Please address that hypothesis.
We have addressed the ontogenetic

Suppl. Material:

Provide specimen numbers in the tables
This has been modified in the manuscript

The wording of the “absent” /” present” state characters need revision. Too many characters states are just “absent” /” present” which is poorly informative. A proper description of the plesiomorphic and apomorphic condition is required. For instance: “195. Surangular and angular divided by posterior extension of the external mandibular fenestra: 0 absent, 1 present”
Could be transformed to something like this: “195. Surangular and angular contact: 0, surangular and angular contact posteriorly; 1, nonexistent because are divided by posterior extension of the external mandibular fenestra.”
The “absent” /” present” may contribute to confusion such as in some cases like ch. 209. “Sternum, distinct lateral xiphoid process posterior to costal margin: 0 absent, 1 present” .
The plesiomorphic condition is the shallow poorly defined (opposed to distinct) process, a medial (opposed to lateral) xiphoid process, or an anterior (opposed to posterior) to costal margin?
Other example: in “214. Surangular, distinct groove on dorsal surface: 0 present; 1 absent” . The apomorphic condition is a shallow groove or in position other than dorsal?
Please rewrite and clarify the characters with “absent” /” present” states.

Provide the list of synapomorphies (the list of characters suffice) for the main clades.

The character matrix is that of a former analysis Lamanna et al. (2014) that was subsequently modified by Funston and Currie (2016) and so we have used the character codings and descriptions of the latter. Revision of the description of the characters for these theropods is well beyond the scope of this paper. The previous authors matrix was used to only determine the phylogenetic position of Baby Louie.

Reviewer #3 (Remarks to the Author):

Review: H. Pu, P. J. Currie, J. Lu, D. K. Zelenitsky, K. Carpenter, L. Xu, E. Koppelhus, S. Jia, H. Chuang, T. Li, M. Kundrat, C. Shen. A perinate of a giant caenagnathid (Oviraptorosauria: Dinosauria) associated with the largest known dinosaur eggs from the Late Cretaceous of Henan, China

The anatomical description of this very significant specimen is thorough and sufficient to distinguish this individual from other individuals. The manuscript, however, can be improved in two major ways in my opinion. The first is relatively straight forward: the use of an updated source matrix for the phylogenetic analysis. The second is more problematic: is it in fact a reasonable action to create a new taxon name for a perinate individual, given that it may well be difficult to assess membership in this taxon in ontogenetically-older specimens. I address each of these in turn.

PHYLOGENETIC ANALYSIS: For purposes of the phylogenetic analysis, the recent paper by Funston & Currie (2016) describing Apatoraptor (ref. 59 in the bibliography) provides an updated version of the source matrix used in the present manuscript. The Funston & Currie version has better resolution in both Oviraptoridae and Caenagnathidae than earlier versions, and so might help break up the massive polytomies found in the present study. Although the manuscript says that this matrix was used, Apatoraptor is absent in Figure 5, which leads me to think that it is actually an earlier version of the matrix that was used for this figure. (The matrix in the supplementary data does contain the new Canadian form, so this is the version that should be used.)

We have used the updated matrix as suggested for our phylogenetic analysis and made this evident in the manuscript.

Additionally, should the new analysis (incorporating Beibeilong and the new characters mentioned in this present manuscript) have the unresolved polytomies

for derived caenagnathids and oviraptorids, I would recommend providing additional consensus tree techniques (e. g., majority-rule; combinable-component, etc.). This could be used to determine if there truly is no structure within these masses of taxa, or instead if there is a shared structure in the most parsimonious trees but one or more rogue taxa have multiple different positions, resulting in strict consensus in which the real structure is obscured. *Use of the updated matrix and methods as suggested has improved the resolution of the phylogenetic tree.*

SHOULD A NEW TAXON BE ERECTED FROM A PERINATE? I can understand and appreciate the desire to assign a taxon name to this relatively complete specimen. And, given that no Gaogou Formation large-bodied caenagnathid genus and species is available to which ‘Baby Louie’ can be referred, a new name would need to be created to accommodate it. So there is nothing wrong per se in erecting “Beibeilong sinensis” for this specimen’s reception.

However, this new name might be quite difficult to apply in future discoveries of post-nestling individuals (especially subadults and adults) of the same taxon, as we may lack direct testable observations to refer said individual to the holotype’s identity. That is because this extremely young individual may lack the morphology of a later ‘ontgimorph’ of the same taxon. (For a comparison, it might be difficult to identify the autapomorphies of Alligator mississippiensis adults in a perinate individual). This is especially likely for issues of proportions and shapes of bones, which are likely to change as an animal grows from the size of a turkey to that of however large Beibeilong became.

Furthermore, the differential diagnosis for Beibeilong may not be applicable to distinguish it from other oviraptorosaur taxa which are known almost strictly from adult or subadult individuals. (On the other hand, it would be a useful set of observations to compare to those oviraptorosaurs known from perinates, just as we could distinguish hatchling Alligator mississippiensis from hatchling Crocodylus niloticus.) Given we do not have good ontogenetic series of any oviraptorosaur taxa, we do not yet have a good prediction of the likely trajectory of changes of shape and proportion of individual skull bones (for instance) to infer a likely shape for the adult Beibeilong skull.

In summary, I am not against the erection of the new name, and would not say it is an inappropriate move. However, I foresee that it might cause difficulty in the future (particularly if two sympatric adult caenagnathid taxa are found in the Gaogou). However, the authors might consider simply referring to the specimen as (alternatively) “Baby Louie” and HGM 41HIII1219, and indicate that they are awaiting discovering a hypodigm including more mature individuals before naming

the taxon.

*There is a distinct suite of characters not present in other oviraptorosaurs that we use to define the taxon, and we have also carefully considered those characters that could be ontogenetically-variable when diagnosing the species. Furthermore, only one other species of giant oviraptorosaur is known, and that specimen is from a different geological stage (Senonian) and a different region of China (northern) than the specimen we describe. Finally, there is precedence in the literature where dinosaur species are named based on very immature specimens, for example, *Microvenator celer*, *Mussaurus patagonicus*, and *Yulong mini*.*

ADDITIONAL COMMENTS: A very useful aspect of this study (beyond the description of this important specimen) is the recognition of new suites of characters that unite Caenagnathoidea and Caenagnathidae.

Also, an interpretive line drawing or labeled photograph of the whole specimen might be very useful. (Ideally, a 3D CT scan of the whole would be even more useful: perhaps these are planned for a future study.)

A new figure has been added in order to show where the eggs are located and positioned. CT scans were unfruitful likely due to the high iron content of the matrix.

Specific Comments and Corrections

p. 2, l. 4 Ken Carpenter' s name is shown inverted. **This was modified in the manuscript.**

p. 5, ll. 94-94 Technically, the Greek (and later Latin) word is Sina; this root word elements become Sin- or Sino- depending on the particular suffix. So this sentence might better read: "Sinensis", Latin for "coming from China". **This was modified in the manuscript.**

p. 24, l. 521 Italicize *Chirostenotes pergracilis* **This was modified in the manuscript.**

p. 26, l. 571 Italicize *Macroelongatoolithus* **This was modified in the manuscript.**

REVIEWERS' COMMENTS:

Reviewer #3 (Remarks to the Author):

This corrected version addresses the points of me and the other reviewers. At this stage I see no problems with publishing the manuscript as it currently stands.